# Bacterial viability in the built environment of the home

Joy Xie ◉°, Ellen M. Acosta ◉°, Zemer Gitai ◉*

Department of Molecular Biology, Princeton University, Princeton, NJ, United States of America

° These authors contributed equally to this work.
* zgitai@princeton.edu

**Data Availability Statement:** All relevant data are within the paper and its Supporting Information files.

**Funding:** Funding was provided in part by NIH (DP1AI124669: Z.G., E.M.A., and J.X., T32 GM007388: E.M.A.). Research reported in this

## Abstract

The built environment (BE) consists of human-made structures and, much like living organisms, is colonized by bacteria that make up the BE microbiome. The BE microbiome can potentially affect human health because of the constant proximity of these bacteria to humans. This has led to increasing public concern of whether the bacteria in the BE are harmful. Previous studies have used approaches based on DNA sequencing to assess the composition of the BE microbiome. However, the extent to which the bacterial DNA in the BE represents viable bacterial cells that could infect human hosts remains unknown. To address this open question we used both culture-based and culture-independent molecular methods to profile bacterial viability of the microbiomes from several BE sites. As part of an undergraduate-led project, we found that the vast majority of the bacterial DNA from the BE is not associated with viable bacteria, suggesting that most bacteria in the BE are dead. To begin to understand the determinants of bacterial viability in the BE we used mock bacterial communities to investigate the effects of temperature, relative humidity, and human interaction on bacterial viability. We found that relative humidity, temperature, and surface material did not have statistically significant effects on BE microbiome viability, but environmental exposure decreased bacterial viability. These results update our conception of the BE microbiome and begin to define the factors that affect BE microbiome viability.

## Introduction

The built environment (BE) consists of human-made structures such as homes, vehicles, work-places, or schools [1]. Like living organisms, the built environment is home to a vast repertoire of microbial life, called the "microbiome," [2]. The BE microbiome and its potential effects on humans has become an increasingly important area of interest as humans spend more time indoors, especially in cities. As of 2020, 56.2% of the world population are living in urban areas with 83% of the U.S. population living in cities, a large jump from 68% in 1950 [3, 4]. This transition to urbanization has caused most people in developed countries to spend 90% of their time indoors [5]. Living indoors was intended to protect people from the harsh factors of the natural environment, including harmful microorganisms often associated with animals or untreated water supplies [5]. But increased association with the BE might also introduce new risks from increased exposure to the BE microbiome.

publication was also supported by the National Center for Advancing Translational Sciences of the National Institutes of Health under Award Number TL1TR003019 (E.M.A.). Additional funding provided by The Evnin '62 Senior Thesis Fund (J. X.) and by the Schmidt Transformative Technology Fund (E.M.A.).

**Competing interests:** The authors have declared that no competing interests exist.

The BE has its own diverse microbiome that is markedly different from the natural environment [5]. This has at times raised public concern when news outlets publish articles that proclaim "These 3 kinds of deadly bacteria are probably in your bathroom right now" or "The NYC Subway is Filled with Bacteria and DNA from Unidentifiable Organisms" [6, 7]. These articles alarm readers when they claim that samples from common BE surfaces contain bacteria associated with the bubonic plague and anthrax or unidentifiable bacteria that have unknown potential to harm people [6, 7]. Previous studies have dispelled some of the alarming claims by demonstrating that the majority of the bacteria in the BE are derived from the human microbiome [8–15]. However, an important unanswered question concerns whether the DNA isolated in traditional BE microbiome sequencing studies reflect DNA from bacteria that are viable and therefore capable of directly colonizing humans and affecting their health (as opposed to indirectly affecting human health by representing a source of potentially harmful DNA sequences that could be taken up by other bacteria via horizontal gene transfer). Multiple sequencing studies have also investigated the environmental factors that alter the BE microbial composition [16–20], but there is little understanding of how external factors affect bacterial viability.

To address these open questions, this study investigated the viability and dynamics of bacteria in the BE through the use of molecular and culture-based methods. We profiled BE microbiome viability in a typical home BE and assessed viability in an isolated environment controlled for environmental factors. We established that surfaces in the home BE have low viability, and we identified factors such as environmental exposure that significantly affected bacterial viability in the BE.

## Materials and methods

### Sample collection, propidium monoazide (PMA) treatment, and DNA isolation

Samples were collected by swabbing surfaces in undergraduate student dorms at Princeton University with sterile swabs and resuspending the swabs in 1500μl of phosphate buffered saline (PBS). Once samples were obtained, each sample was split in half with 500 μl each in two 1.5 mL microcentrifuge tubes. Next, 1.25 μl of cell-impermeable small molecule propidium monoazide (PMA) was added to one of the two tubes at a final concentration of 50 μM. PMA was used to assess the overall viability of the BE microbiome because it binds to dead bacteria and inhibits PCR amplification (Fig 1) [21]. Then, both halves of the sample were incubated in the dark for 10 minutes at room temperature. To activate PMA, all tubes were exposed to light using the PMA-Lite™ LED Photolysis Device for 20 minutes. DNA isolation for all samples was performed using the DNeasy PowerSoil Pro Kit (Qiagen).

### Droplet digital PCR (ddPCR)

To quantify bacterial viability in the BE, the Bio Rad QX200 AutoDG Droplet Digital PCR System was used. For one reaction, the master mix contained 12.5 μl of 2x Q200 ddPCR Eva green Supermix, 5 μl of universal 16S qPCR primers at 10 nM concentrations, 3.5 μl of DNase free water. 21 μl of the master mix was combined with 4 μl of the sample (see Table 1 for primer sequences). Reaction mixtures were then pipetted into a sterile ddPCR 96 well plate and heat-sealed using the PX1 PCR Plate Sealer. The plate was then loaded into the Qx200 Automated Droplet Generator. After droplets were generated, the plate was heat sealed again and loaded into a C100 Touch Thermal Cycler for PCR. There was a pre-step of 95° C for 5 minutes, then 40 rounds of amplification at 60° C, 1 minute for extensions, and a final hold

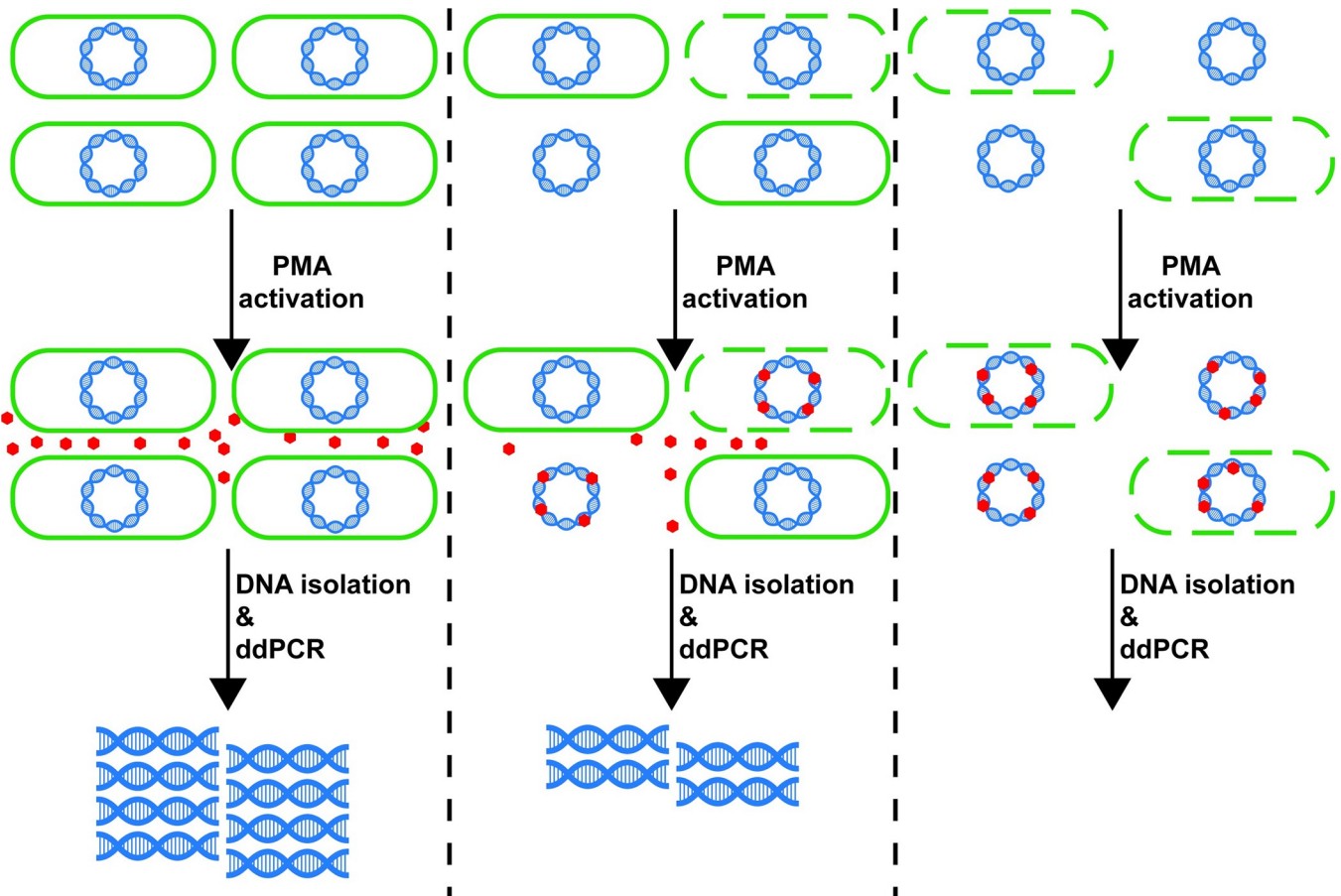

**Fig 1. Evaluating bacterial viability using PMA-ddPCR.** PMA can enter cells with damaged membranes or can encounter extracellular DNA, where it binds irreversibly to double stranded DNA and inhibits detection by ddPCR. PMA is excluded from viable cells. This allows us to calculate bacterial viability through quantifying the ratio of DNA with and without PMA using ddPCR.

temperature at 12° C. For quantification, the plate was then loaded into the QX200 Droplet Reader. The Quantasoft software recorded the data, which was then exported to Microsoft Excel for cleanup and analysis. Viability score was calculated for paired samples of with and without PMA treatment using the formula: Copies per 20 μl with PMA/ Copies per 20 μl without PMA [22].

## Culturing

To compare viability scores determined through PMA-ddPCR to culturable bacteria, samples used for PMA-ddPCR were cultured prior to the addition of PMA. Depending on the initial bacterial concentration in the sample, appropriate 10-fold dilutions were made. 100μl of each

**Table 1. Primers used for ddPCR and PCR.**

| Name | Description | Sequence (5'-3') | Reference |
|---|---|---|---|
| ddPCR forward primer (FP) | Universal bacterial 16S qPCR FP | TCCTACGGGAGGCAGCAGT | [22, 25] |
| ddPCR reverse primer (RP) | Universal bacterial 16S qPCR RP | GGACTACCAGGGTATCTAATCCTGTT | [22, 25] |
| D88 | FP for PCR | GAGAGTTTGATYMTGGCTCAG | [23] |
| E94 | RP for PCR | GAAGGAGGTGWTCCARCCGCA | [23] |

dilution was plated onto blood agar plates containing 5% sheep blood in tryptic soy agar (VWR International). Blood agar was selected because it is an enriched, non-selective media on which a wide variety of bacterial species can grow, including fastidious microorganisms. To ensure that slow-growing colonies were accounted for, plates were examined under a dissecting microscope that allowed even very small colonies to be visualized. Sterile glass beads were added to the plates, and the plates were shaken to spread the dilutions evenly across the plates. The beads were then dispensed, and the plates were placed in a 37˚ C warm room to culture overnight. To preserve cultured bacteria for potential use in a mock community, we grew single colonies in Brain-Heart Infusion (Difco) or Tryptic Soy Broth (Difco) and froze them down by creating a mixture of 300μl of 50% glycerol and 900μl of bacteria in a tube and placing them in a -80˚ C freezer.

## Counting colony-forming unit (CFU)/mL

After bacterial colonies grew overnight on the blood agar plates, CFUs were counted using a colony-counter pen (Scienceware, Bel-Art Products), and the CFUs were recorded for each plate. CFU/ml was calculated by using this formula: (CFU counted x dilution factor)/ mL plated

## Sanger sequencing

To determine the species of culturable bacteria in the BE, bacterial colonies that grew overnight on the blood agar plates were sequenced. To ensure that all cultured bacteria were identified, 3 colonies of each colony morphology on a given plate were picked with sterile toothpicks and placed in 20μl of distilled water in sterile 200 μl PCR tube strips. The abundance of each type of colony morphology was recorded in order to approximate relative abundance after species identification. 1.5 KB of the 16S gene of each colony was amplified using primers D88 and E94 to ensure accurate species identification [23]. Each reaction was PCR-purified and sent to Azenta for Sanger Sequencing and bacterial species were identified using BLAST [24]. Additional information can be found in S3 Table.

## Mock community

We created a mock community consisting of the top five most frequently occurring bacteria in the home BE (S1 Table). We grew overnight liquid cultures of each species. We used a pipette tip to take bacteria from the cultures that we froze from swabbing surface in the home BE and placed them in 5mL of Brain Heart Infusion (Difco). They were then placed in a shaker at 37˚ C for 12 hours. Our minimal mock community consisted of 30% *Acinetobacter ursingii*, 23% *Stenotrophomonas maltophilia*, 18% *Staphylococcus epidermidis* and *Staphylococcus hominis*, and 11% *Micrococcus yunnanesis*. Species included in the mock community came from single colonies isolated during our environmental sampling.

To examine the impact of external factors on bacterial viability, we plated the mock community on surfaces in a controlled environment where most conditions were fixed except the variable we were examining (S1 Fig). We sterilized each surface with 70% ethanol and placed it into a larger lidded sterilized box (also sterilized using 70% ethanol) to prevent human interaction with the surface through touch or the air. Five different surface types were used: polystyrene foam (Antique Ceilings, Inc.), unfinished birch plywood (Coff), stainless steel (K & S Precision Metals), polypropylene plastic (pipette tip boxes, USA Scientific), and glass (microscope slides, Fisher Scientific). We then sectioned each surface into 6 squares of 2 by 2 inches (5.08 by 5.08 cm) and plated 20μl of the mock community onto each square. We swabbed a square at 6 timepoints over one week: 0 hours, 5 hours, 1 day, 3 days, 5 days, and 7 days. Each

square was only swabbed once for each timepoint. These timepoints were selected to see both short term and long-term effects of surface material on viability. During each sampling time-point, the box was opened under sterile conditions. We then quantified viability with PMA-ddPCR.

For the relative humidity timepoint experiments, we used the same overall method with modifications to the setup (S2 Fig). To maintain 99% relative humidity, 100mL of distilled water was placed in the sterile environment. For 60% relative humidity, 100mL of distilled water with 70g NaCl was placed in the sterile environment. Salt binds to water and prevents water vapor formation, thereby maintaining a lower RH [26]. We confirmed that RH levels were maintained by using humidity monitors (ThermoPro TP50 Digital Hygrometer).

## Results

### The bacterial DNA on most surfaces in the home-built environment is associated with low bacterial viability

Many of the efforts towards characterizing the built environment microbiome have relied on bacterial 16S rRNA gene sequencing to identify bacterial species [8–15]. While this is a power-ful tool and has been essential for microbiome research, 16S rRNA gene sequencing does not differentiate between DNA from live and dead bacterial cells. This distinction is important when evaluating bacterial communities such as the BE microbiome because the presence of bacterial DNA does not necessarily equate to the presence of viable microorganisms, nor does the presence of DNA indicate infection potential. Therefore, to evaluate bacterial viability, we utilized a molecular method combining propidium monoazide (PMA) and droplet digital PCR (ddPCR) to differentiate between DNA from viable cells and DNA from nonviable cells (Fig 1) [22]. PMA is a membrane-impermeable molecule that upon photoactivation interacts with DNA in a manner that inhibits PCR amplification [27–29]. Because PMA cannot pene-trate live bacteria, it should specifically interact with DNA from dead or damaged cells (or extracellular DNA). To calculate what fraction of bacterial DNA originated from live or viable cells, we divided each sample into two paired aliquots of equal volumes and quantified the ratio of amplifiable DNA abundance in each paired samples with and without PMA treatment by ddPCR. This process enabled us to calculate a viability score for any given microbiome sam-ple ranging from 0 (no DNA was associated with viable cells) to 1 (all DNA was associated with viable cells).

To survey viability scores of microbiome samples from the BE, we first profiled various sur-faces in a typical home (S1 Table). Using sterile swabs, we sampled surfaces included in the National Sanitation Foundation's "top germiest" places in a home: toothbrush, door handle, cell phone, kitchen sink, sponge, kitchen faucet, and the inside of a fridge [30]. We then used the PMA-ddPCR approach to quantify bacterial viability using universal primers specific for the highly conserved bacterial 16S rRNA gene. Using PMA-ddPCR also allowed us to account for viable but nonculturable bacteria, which would otherwise be difficult to assess through tra-ditional methods such as culturing. We found that most surfaces sampled had low viability scores, with 14 of the 17 surfaces (82%) exhibiting less than 20% viability (Fig 2A). In compari-son, an exponentially-growing culture of *S. epidermidis*, a common bacterial species in the built environment, had a viability score of 0.988 (Fig 2A). As another point of comparison, in a separate study our lab used the same approach to show that the majority of the bacteria human and mouse fecal samples are viable (viability scores equal to 0.66 and 0.98) [22]. These results showing relatively low viability in the BE confirm recent cleanroom studies that also used PMA and quantified that microbial viability in cleanroom environments is in a range of 1% to 7% [21, 31].

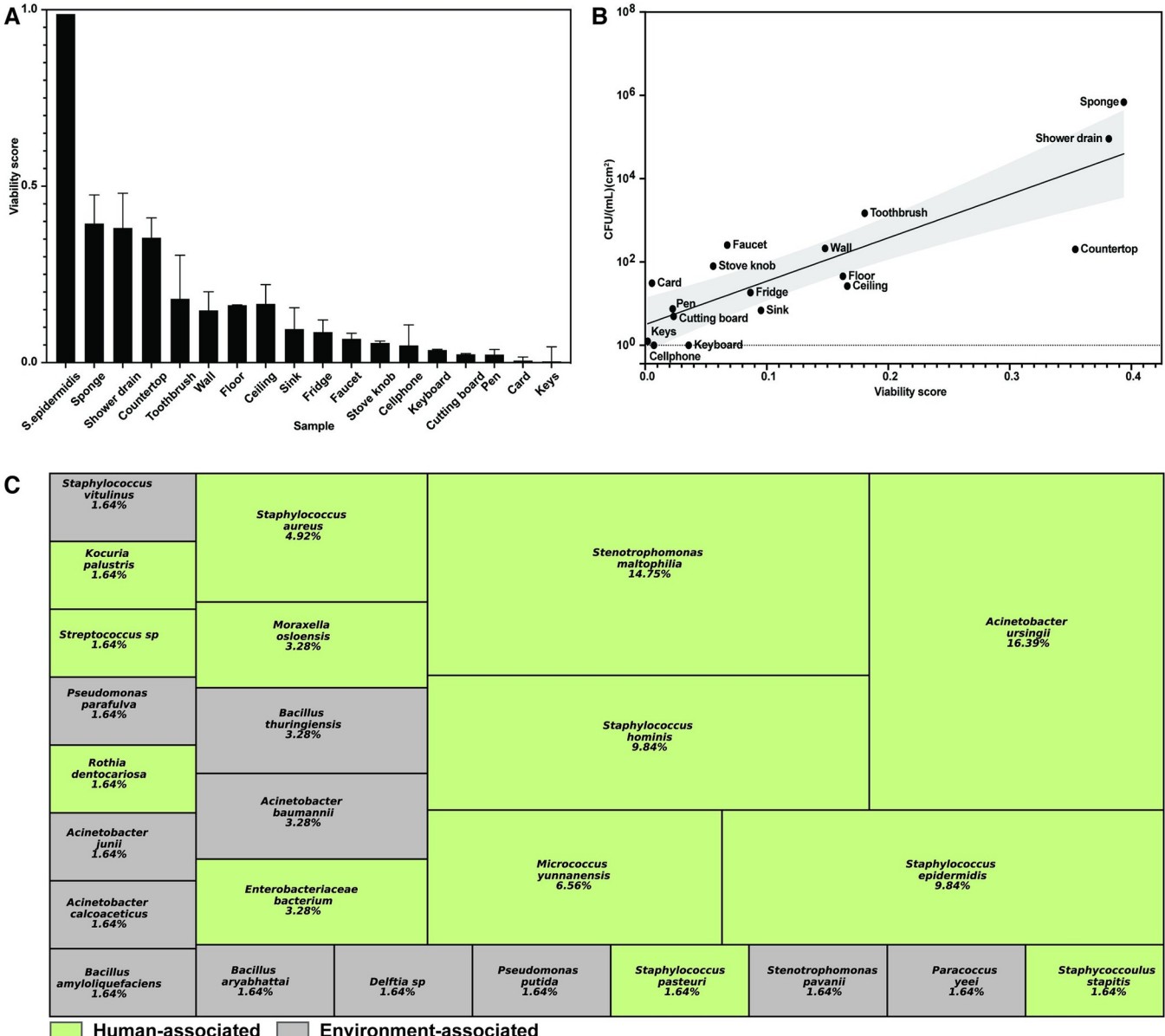

**Fig 2. Viability Scores of varying home BE surfaces are low.** (A) Viability Scores of varying home BE surfaces. Error bars represent standard deviation. (B) CFU/(ml)(cm$^2$) of varying home BE surfaces. (C) CFU/(mL)(cm$^2$) and viability scores are correlated. Shaded gray region represents the 95% confidence interval. (D) Bacterial species identified during home BE profiling. Green indicates human-associated bacterial species, gray indicates environmental-associated bacterial species. 77.05% of the species identified were human-associated. The relative size of each box reflects the relative frequency with which that species was identified.

Despite the potential presence of nonculturable cells, culturing-based methods can offer powerful insight when combined with molecular approaches such as PMA-ddPCR. Therefore, we also cultured the same samples as those used for PMA-ddPCR. These cultures enabled us to quantify the population of viable and culturable bacteria in the BE and to explore the relationship between culturability and PMA-ddPCR viability at different environmental locations. We found that quantifying culturable bacteria using classical CFU plating generally correlated well with the PMA-ddPCR viability scores, as surfaces with higher viability scores had higher CFU counts and surfaces with lower viability scores had lower CFU counts (Fig 2B). While

most sample sites were within the 95% confidence interval for the correlation between CFU and PMA-ddPCR viability predicted by linear regression, some sample sites. Weak correlation between CFU and PMA-ddPCR viability could reflect an environment where most of the bacteria are viable but are scarce or nonculturable, resulting in lower than expected correlations as seen on countertops in our data. Alternatively, an environment with many culturable bacteria but even more non-viable bacteria may have a high number of CFU and a low viability score, resulting in higher than expected correlations as seen on faucets in our data.

Performing CFU plating also enabled us to identify the bacterial species present in the home BE microbiome. By performing colony PCR with primers specific for the bacterial 16S rRNA gene and sequencing the PCR products, we determined the species identity of 61 randomly-selected BE microbiome culture isolates (see methods for details). We found that most of the bacterial species (77.05%) present in the home BE were human-associated bacterial species that are commonly found in human skin, gut, oral cavity, or respiratory tract (Fig 2C and S1 Table). The other 22.95% of bacterial species are commonly found in environments such as soil and water. Some environmental-associated bacterial species, such as *Acinetobacter baumannii*, are also opportunistic human pathogens, but because they are not human commensals, they were categorized as environmental-associated.

## Temperature, relative humidity, and surface material have little effect on bacterial viability

Our PMA-ddPCR analysis demonstrated that there are significant differences in bacterial viability at different sites in the home, as we identified viability scores that ranged from 0.002 (on a key) to 0.39 (on a sponge). We hypothesized that these differences in viability could be driven by differences in local environmental factors like moisture, temperature, or surface composition.

To understand how specific environmental factors affect BE bacterial viability in a controlled manner, we first assembled a simplified mock community of bacterial species from our cultured environmental isolates (Fig 2C and S1 Table). Using only the top five most frequently isolated bacterial species, our mock community consisted of 30% *Acinetobacter ursingii*, 23% *Stenotrophomonas maltophilia*, 18% *Staphylococcus epidermidis* and *Staphylococcus hominis*, and 11% *Micrococcus yunnanesis*. We applied this mock community to sterilized plastic surfaces contained in a controlled environment in which we could adjust the water content and temperature. We then monitored bacterial viability scores over one week, sampling at 0 hours (input culture), 4 hours, 24 hours, 72 hours, 120 hours, and 168 hours after application of the mock communities. To isolate the effect of water content level, we examined viability scores at relative humidity (RH) levels of 60% and 99%. To maintain 60% RH, we included a container filled with a saturated salt solution in the controlled environment, and to maintain 99% RH we used a container of water [26].

We first examined the effect of environmental water content at room temperature, or 22˚C, as this would best represent the home BE (Fig 3A). We found that bacterial viability at 99% RH was slightly elevated (0.39), but not significantly different (p = 0.842) from 60% RH (0.22) (Fig 3A and S2 Table). To evaluate the effect of different relative humidity levels at different temperatures, we also measured bacterial viability over one week at 60% and 99% RH at 4˚C and 37˚C. As with 22˚C, there was no statistically-significant difference in viability between the two RH levels in either high or low temperature environments (p = 0.374 and 0.985, Fig 3A). At 4˚C and 22˚C, there was also no statistically-significant difference in CFU (p = 0.129 and 0.201), but there was an increase in CFU at 37˚C with 99% RH (p = 0.044) (Fig 3B). Additionally, we evaluated bacterial viability over one week at 4˚C, 22˚C, and 37˚C on surfaces exposed to the ambient environmental RH levels, and found no difference between the three

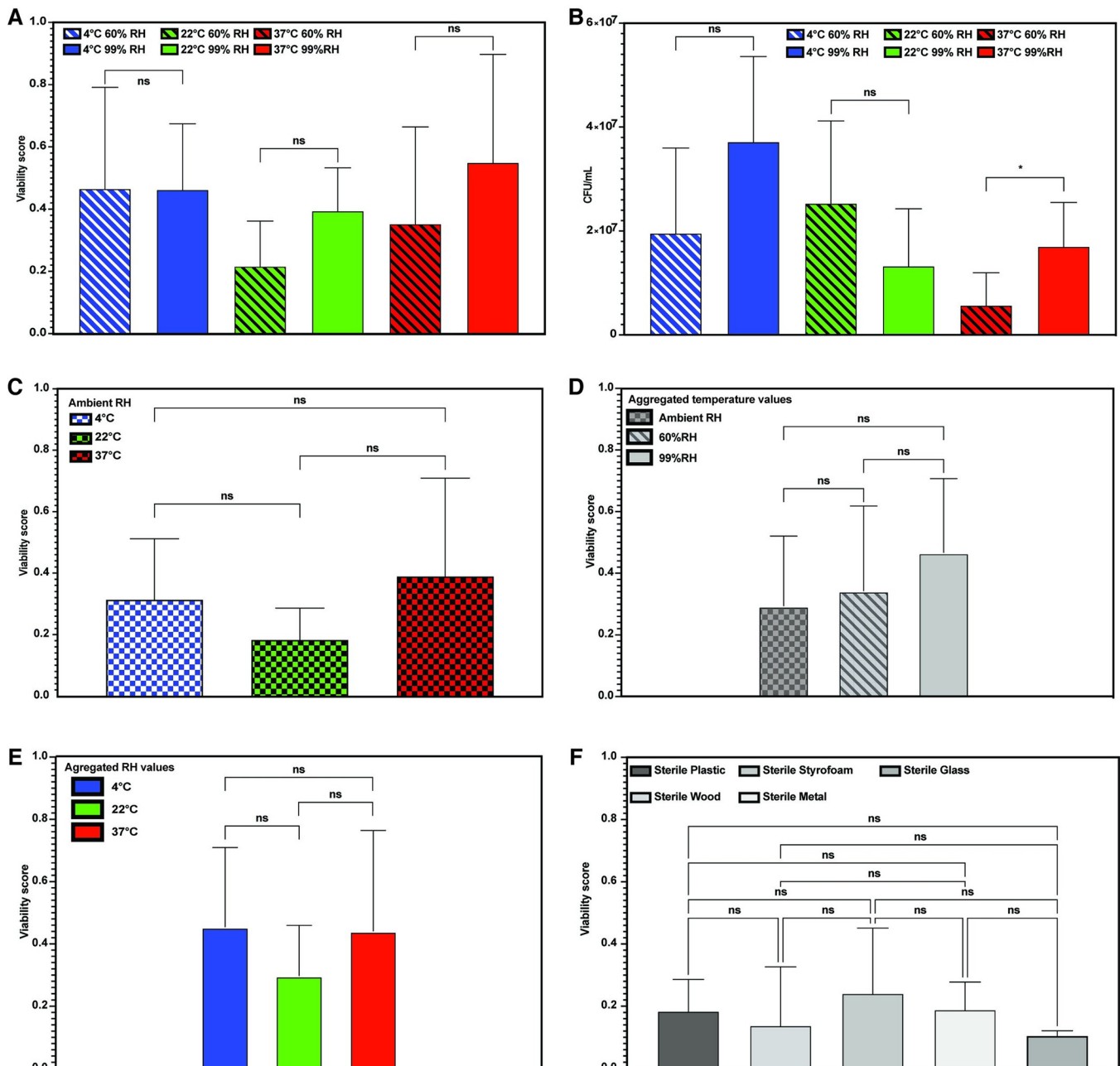

**Fig 3. The effect of temperature and relative humidity on the BE microbiome.** (A-C) Average viability of bacterial mock communities plated on plastic surfaces at 60% RH (striped bars) or 99% RH (solid bars) and sampled over 7 days at 22˚C (green), 4˚C (blue), and 37˚C (red). Averages include all timepoints after the input at $T_0$. (D) Average viability of bacterial mock communities plated on plastic surfaces at ambient RH (hatched bars) sampled over 7 days at 22˚C (green), 4˚C (blue), and 37˚C (red). (E) Average viability of bacterial mock communities at all temperatures (22˚C, 4˚C, and 37˚C) for ambient RH (hatched bar), 60% RH (striped bar), and 99% RH (solid bar). (F) Average viability of bacterial mock communities at all RH levels (ambient, 60%, and 99%) for 4˚C (blue), 22˚C (green), and 37˚C (red). Error bars represent standard deviation. * = p < 0.05. Student's two-tailed T test used for A and B. Tukey multiple comparisons test used for C-F. See S2 Table for full list of p values.

temperatures (p = 0.642, 0.350, and 0.860) (Fig 3C). Aggregating all data taken from different temperatures but the same RH level or different RH levels but the same temperature also showed no difference in bacterial viability, again suggesting that neither temperature nor water content level is driving bacterial viability in the home BE (Fig 3D and 3E).

Another factor that could influence BE microbiome viability is the ability of bacteria to survive on different surfaces found in the home. To assess the effects of different surface materials on viability we applied our mock community of BE bacterial species to 5 different surface materials (plastic, wood, glass, Styrofoam, and metal) and assessed bacterial viability over the course of 7 days (Fig 3F and S2 Table). We found that there was no significant difference in viability across the five materials tested, suggesting that surface material properties could also not fully account for the differences in BE microbiome viability.

## Human interaction decreases bacterial viability in the BE

While the effects of water content and temperature were minimal, there are many other environmental factors that may impact bacterial viability in the BE. For example, if most of the bacteria in the home BE are human-associated species, then surfaces with high human interaction might have higher bacterial viability as a result of more frequent transfer from humans. Alternatively, recent work from our group has shown that most bacteria associated with the human skin microbiome are non-viable, suggesting that human-to-BE bacterial transfer may result in low bacterial viability [22].

Using the home BE profiling data, we first categorized the surfaces into high contact (Sponge, Toothbrush, Faucet, Fridge, Keyboard, Pen, Credit card, Cellphone, and Keys) and low contact (Shower drain, Countertop, Sink, Wall, Ceiling, Floor, Cutting board, and Stove knob) based on how frequently they came into physical contact with humans. We found that high human contact surfaces had lower average viability (0.094), while low human contact surfaces had higher average viability (0.173), but this difference was not statistically significant (p = 0.219 Fig 4A). Because these data were taken from profiling many different surfaces in the home BE, an environment that is difficult to control, we compared bacterial viability on each of the 5 surfaces examined in Fig 3F in a sterile environment protected from human contact to an exposed (uncovered) environment on a laboratory benchtop. For each experiment we assessed bacterial viability over the course of 7 days.

Consistent with our results from a sterile environment, we found that surface material type had no impact on bacterial viability in the exposed environment. However, cultures placed in the sterile, enclosed environment had significantly higher average viability (0.17) than those placed in the non-sterile, exposed environment (0.10) (p = 0.0296, Fig 4B and S2 Table). Furthermore, the average viability of surfaces in the non-sterile environment decreased by 90% over 7 days, while the average viability of surfaces in the sterile environment decreased by just 45% (Fig 4C).

Together, these data suggest that in contrast to the expectation that surfaces with the most exposure might accumulate the most bacteria, environmental exposure decreases bacterial viability in the BE. Most bacteria in the home BE are human-associated species, suggesting that they may be transferred to the BE from human skin, which is dominated by a microbiome with low viability [22]. Increased human interaction may also lead to increased exposure to antimicrobial agents found in the environment, which could also contribute to the difference in bacterial viability in our controlled experimental setup.

## Discussion

Overall, we found that the built environment has low bacterial viability and that temperature, humidity, and surface material do not significantly affect that viability while environmental exposure does. Using the PMA-ddPCR approach, we were able to evaluate bacterial viability of culturable and nonculturable bacteria and showed that viability correlated well with CFU/mL in the home BE. These results confirm recent studies and expand on their findings by

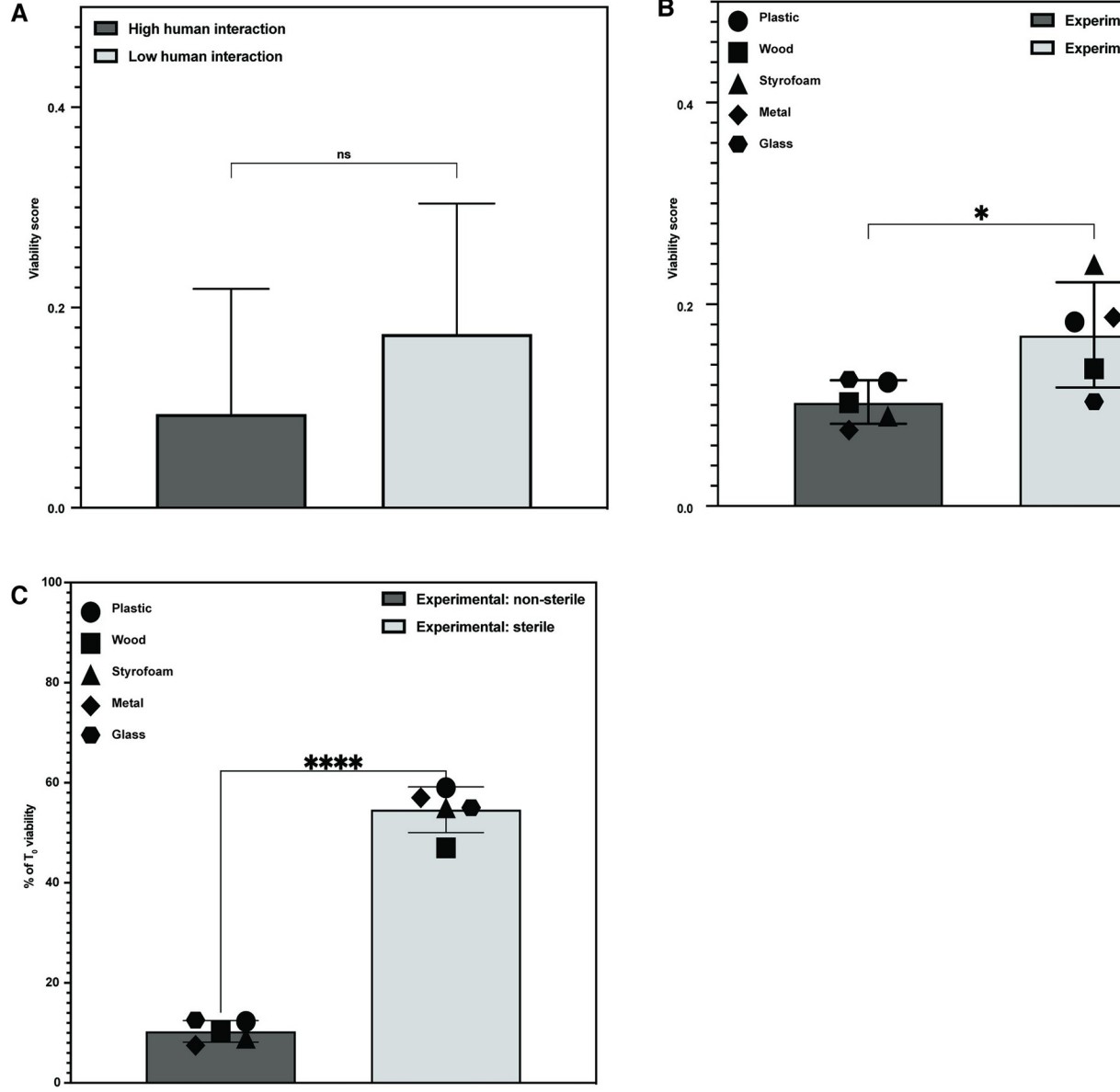

**Fig 4. The effects of interaction on bacterial viability.** (A) Comparison of average viability scores based on human interaction. Average viability score for high human contact surfaces (Sponge, Toothbrush, Faucet, Fridge, Keyboard, Pen, Credit card, Cellphone, and Keys) was 0.094. Average viability score for low human contact surfaces was (Shower drain, Countertop, Sink, Wall, Ceiling, Floor, Cutting board, and Stove knob) 0.173. Bars represent standard deviation. (B) Average viability after $T_0$ of bacterial mock communities plated on different surface material types at ambient RH and 22°C sampled over 7 days in either an uncovered, exposed environment (0.10) or a sterile, protected environment (0.17). (C) Average viability as a percent of input ($T_0$) viability of bacterial mock communities plated on different surface material types. The average bacterial viability after $T_0$ as was 10% of the input for the non-sterile experimental setup and 54% for the sterile experimental setup. Statistical tests are Student's two-tailed T-test. * = p<0.05, **** = p<0.0001. Bars represent standard deviation.

demonstrating the specific effects that surface material, humidity, temperature, and human interaction have on bacterial viability [13, 21]. Together, these results indicate that most bacteria in the BE are nonviable, suggesting that frequent sterilization with strong cleaning chemicals in a typical home may not be as necessary to reduce transmission of harmful bacteria. Furthermore, a majority of culturable bacteria in the home BE are known to be human-associated, suggesting that human microbiomes act as reservoirs for populating the home BE.

Results from both profiling the home BE and following the timeline of our mock community experiments suggest that water content level, temperature, and surface composition have little effect on bacterial viability, while environmental exposure negatively impacts viability in the BE. We also observed that high human contact surfaces such as keyboard had lower viability than low human contact surfaces such as walls. These results suggest that human interaction may drive overall BE microbiome dynamics by decreasing viability. We hypothesize that environmental exposure lowers viability by either contributing more dead bacteria than viable ones to the BE or by delivering antimicrobial agents [22, 32]. Future studies will be needed to differentiate these possibilities, as well as to resolve whether our inability to detect significant effects of other environmental factors is a function of the statistical power of our experiment.

Our findings, along with other studies, suggest that BE microbiome may not pose as big of a threat to human health as initially feared. We found that most bacteria in the BE are nonviable, suggesting that aggressive, frequent sterilization of the home BE may not be productive and could even prove counter-productive as it could lead to the emergence of antimicrobial resistance in the BE [33]. In the future, additional studies will help to determine if these findings hold true across other BE bacterial species and environments.

## Supporting information

**S1 Fig. Timepoint experiment schematic.** Mock community was plated onto surface and placed into sterile environment.
(TIF)

**S2 Fig. Relative humidity timepoint experiment schematic.** Mock community was plated onto surface. Salt and water solution was created to maintain relative humidity in a sterilized environment. Both were placed into a sterilized environment.
(TIF)

**S1 Table. Surface samples and their bacteria species.** A list showing how surfaces were categorized under each location and showing what bacteria species were on each surface.
(DOCX)

**S2 Table. Statistical test information for Figs 3 and 4.** The details for each statistical test performed in this study are included here.
(DOCX)

**S3 Table. Sanger sequencing and BLAST results.** Representative sequences for each bacterial species identified in this study and details from BLAST, including query, length of query, query coverage, E-value, and percent identity.
(XLSX)

## Author Contributions

**Conceptualization:** Joy Xie, Ellen M. Acosta, Zemer Gitai.

**Data curation:** Joy Xie.

**Formal analysis:** Joy Xie, Ellen M. Acosta.

**Funding acquisition:** Zemer Gitai.

**Investigation:** Joy Xie.

**Methodology:** Joy Xie, Ellen M. Acosta, Zemer Gitai.

**Resources:** Zemer Gitai.

**Supervision:** Ellen M. Acosta, Zemer Gitai.

**Visualization:** Joy Xie, Ellen M. Acosta.

**Writing – original draft:** Joy Xie, Ellen M. Acosta, Zemer Gitai.

**Writing – review & editing:** Joy Xie, Ellen M. Acosta, Zemer Gitai.

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
