## [Decision Letter · Decision Letter 0]

19 Apr 2023

PONE-D-23-07442Bacterial viability in the built environmentPLOS ONE

Dear Dr. Acosta,

Thank you for submitting your manuscript to PLOS ONE. After careful consideration, we feel that it has merit but does not fully meet PLOS ONE’s publication criteria as it currently stands. Therefore, we invite you to submit a revised version of the manuscript that addresses the points raised during the review process. Please submit your revised manuscript by Jun 03 2023 11:59PM. If you will need more time than this to complete your revisions, please reply to this message or contact the journal office at plosone@plos.org. Please include the following items when submitting your revised manuscript:A rebuttal letter that responds to each point raised by the academic editor and reviewer(s). You should upload this letter as a separate file labeled 'Response to Reviewers'.A marked-up copy of your manuscript that highlights changes made to the original version. You should upload this as a separate file labeled 'Revised Manuscript with Track Changes'.An unmarked version of your revised paper without tracked changes. You should upload this as a separate file labeled 'Manuscript'.If applicable, we recommend that you deposit your laboratory protocols in protocols.io to enhance the reproducibility of your results. Protocols.io assigns your protocol its own identifier (DOI) so that it can be cited independently in the future. For instructions see: https://journals.plos.org/plosone/s/submission-guidelines#loc-laboratory-protocols. Additionally, PLOS ONE offers an option for publishing peer-reviewed Lab Protocol articles, which describe protocols hosted on protocols.io. Read more information on sharing protocols at https://plos.org/protocols?utm_medium=editorial-email&utm_source=authorletters&utm_campaign=protocols.

We look forward to receiving your revised manuscript.

Kind regards,

Rajeev Singh

Academic Editor

PLOS ONE

Journal Requirements:

Reviewers' comments:

Reviewer's Responses to Questions

**Comments to the Author**

1. Is the manuscript technically sound, and do the data support the conclusions?

Reviewer #1: Yes

Reviewer #2: Partly

Reviewer #3: Yes

2. Has the statistical analysis been performed appropriately and rigorously? 

Reviewer #1: Yes

Reviewer #2: Yes

Reviewer #3: Yes

3. Have the authors made all data underlying the findings in their manuscript fully available?

Reviewer #1: Yes

Reviewer #2: No

Reviewer #3: Yes

4. Is the manuscript presented in an intelligible fashion and written in standard English?

Reviewer #1: Yes

Reviewer #2: Yes

Reviewer #3: Yes

5. Review Comments to the Author

Reviewer #1: Excellent contribution about daily micro biome environment and can apply for hygiene and health care.

Only minor points

1.title is not specifically site enough or geographical site.

2. Figure 5 must be improve for more easy and clear for reading and publish.

Regards,

Reviewer #2: This manuscript investigated the viability of bacteria in the built environment using culture-dependent and culture-independent approach. The content of the manuscript fits well with the aim and scope of the journal and should attract a wide readership. Overall quality of the manuscript is good. The manuscript is well written and clearly explained. Therefore, I recommend a minor revision of this manuscript in its current format. Some concerns are listed below.

Materials and methods

In general, this section is well written. However, some parts are lengthy and can be reduced in particular the Sanger sequencing part.

Line 388 For one volume,…. Do the authors mean reaction?

Line 389 DNase

Line 397-399Please add reference for viability score calculation formula

Line 405 Why did the authors rely on the isolation of cultivable bacteria on blood agar? The use of only blood agar and short incubation period led to a bias toward pathogenic bacteria and fast growing bacteria. There is likely that the authors will miss out on the slow growing bacterial species in the built environment. Please discuss this point in the discussion section.

Line 452 I assumed that the authors used BLAST in NCBI, I strongly recommend the use of EzBioCloud website for a more accurate taxonomic assignment of the bacteria. Please add 16S rRNA gene similarity value from BLAST analysis and length of the 16S rRNA gene in Table S1.

Line 461-463Are these 5 bacteria for mocking experiment from culture collection or from your own isolation? Please specify accordingly.

Line 466How did the authors check sterility of each surface?

Line 467 How did the authors sterilize the box and ccheck sterility of it?

How many replicates for each time point? Only one experiment or repeated experiment?

Please give detail of surface types under investigated in this section.

Line 481 70g of what?

Line 481-482Rewrite this sentence “Salt maintains binds to water and prevents water vapor formation, thereby maintaining a lower RH (28).”

Results

I recommend the authors rewrite this section as in its current format the results are mixed with discussion.

There is no information of the accession number of all 16S rRNA gene sequences generated from this study in public database.

The resolution of all figures are not good enough. They are blurred except for Figure 1.

Why there is A in Figure 1?

There is no Figure2D in figure but it was mentioned in the text.

Figure S2 what is the object on the right hand side.

Table S1 Please add the length of the 16S rRNA gene, accession number and 16S rRNA gene similarity value from BLAST analysis.

Table S1 delete the word “strain” and the word “sp” must not be italicized.

Table S2 Column 2 what is “Pannel”? Do the authors mean “panel”?.

Table S2 Column 3, Suggest change the word “comparison” to something else eg. conditions or treatments

Reference section

Please check carefully for errors eg. no journal name+volume+page number for #10, 18, #34 no volume+page number.

Reviewer #3: This is very interesting and timely study. Authors have documented the built environment microbiome availability in different locations. Data were analyzed and presented accurately and informative to the readers. It is a well written study based on properly conducted experiments.

6. PLOS authors have the option to publish the peer review history of their article (what does this mean?). If published, this will include your full peer review and any attached files.

Reviewer #1: No

Reviewer #2: No

Reviewer #3: No

---

## [Author Response · Author response to Decision Letter 0]

26 May 2023

We thank the reviewers for their helpful suggestions and comments. A response to reviewers has been included in this resubmission as a separate attached document.

Additionally, the manuscript has been re-formatted to adhere to the formatting standards of PLOS One.

---

## [Editor Report · Decision Letter 1]

19 Jun 2023

Bacterial viability in the built environment of the home

PONE-D-23-07442R1

Dear Dr. Acosta,

We’re pleased to inform you that your manuscript has been judged scientifically suitable for publication and will be formally accepted for publication once it meets all outstanding technical requirements.

Kind regards,

Rajeev Singh

Academic Editor

PLOS ONE
---

## [Editor Report · Acceptance letter]

5 Jul 2023

PONE-D-23-07442R1 

Bacterial viability in the built environment of the home 

Dear Dr. Acosta:

I'm pleased to inform you that your manuscript has been deemed suitable for publication in PLOS ONE. Congratulations! Your manuscript is now with our production department. 

Kind regards, 

on behalf of

Dr. Rajeev Singh 

Academic Editor

PLOS ONE